# Peptide Receptor Radionuclide Therapy with [^177^Lu]Lu-DOTA-TATE in Patients with Advanced GEP NENS: Present and Future Directions

**DOI:** 10.3390/cancers14030584

**Published:** 2022-01-24

**Authors:** Maria I. del Olmo-García, Stefan Prado-Wohlwend, Pilar Bello, Angel Segura, Juan F. Merino-Torres

**Affiliations:** 1Department of Endocrinology and Nutrition, University and Polytechnic Hospital La Fe, 46026 Valencia, Spain; merino_jfr@gva.es; 2Department of Nuclear Medicine, University and Polytechnic Hospital La Fe, 46026 Valencia, Spain; prado_ste@gva.es (S.P.-W.); bello_pil@gva.es (P.B.); 3Department of Medical Oncology, University and Polytechnic Hospital La Fe, 46026 Valencia, Spain; segura_ang@gva.es

**Keywords:** neuroendocrine neoplasms, PRRT, [^177^Lu]Lu-DOTA-TATE

## Abstract

**Simple Summary:**

Neuroendocrine neoplasms have been usually described as infrequent tumors, but their incidence has been rising over time. [^177^Lu]Lu-DOTA-TATE (PRRT-Lu) was approved by the European Medicines Agency and by the Food and Drug Administration as the first radiopharmaceutical for peptide receptor radionuclide therapy in progressive gastroenteropancreatic NET. PRRT-Lu is considered a therapeutic option in progressive SSTR-positive NETs with homogenous SSTR expression. The NETTER-1 study demonstrated that PRRT-Lu yielded a statistically and clinically significant improvement in PFS as a primary endpoint (HR: 0.18, *p* < 0.0001), as well as a clinical trend towards improvement in OS. These results made scientific societies incorporate PRRT-Lu into their clinical guidelines; however, some questions still remain unanswered.

**Abstract:**

This review article summarizes findings published in the last years on peptide receptor radionuclide therapy in GEP NENs, as well as potential future developments and directions. Unanswered questions remain, such as the following: Which is the correct dose and individual dosimetry? Which is the place for salvage PRRT-Lu? Whicht is the role of PRRT-Lu in the pediatric population? Which is the optimal sequencing of PRRT-Lu in advanced GEP NETs? Which is the place of PRRT-Lu in G3 NENs? These, and future developments such as inclusion new radiopharmaceuticals and combination therapy with different agents, such as radiosensitizers, will be discussed.

## 1. Introduction

Neuroendocrine neoplasms (NENs) are typically described as rare tumors; however, their incidence has been rising over time [1,2]. Neuroendocrine cells are distributed widely throughout the body, and NENs arise from them; therefore, these tumors can appear in organs such as the pancreas, foregut, midgut, hindgut, or bronchial tree, or other unusual locations such as the ovary, cervix, or breast. NENs are classified on the basis of two parameters: cell differentiation and rate of proliferation. The World Health Organization (WHO) classified gastroenteropancreatic NENs (GEP-NENs) accordingly as well differentiated neuroendocrine tumors (NETs) and poorly differentiated neuroendocrine carcinomas (NECs). NENs arising in other localizations, such as bronchial NENs, do not follow this same classification [3]. Within NENs, GEP NETs are the most common subtype, representing more than 70% of the totality [4].

Although complex and heterogeneous, somatostatin receptor (SSTR) overexpression on their cell surface is frequent, and this is the reason why these receptors have been a main concern of study in these tumors [5]. Five Somatostatin G protein-coupled receptors have been identified: SSTR1, SSTR2, SSTR3, SSTR4, and SSTR5. SSTRs are widely distributed in healthy tissues, with distinct expression throughout the body. Tumor cells and peritumoral vessels express different SSTR subtypes whose density depends on the type of tumors. SSTR expression may vary within the different tumors and, in particular, in NETs [6,7,8] (Table 1 and Table 2).

Among the SSTR subtypes, SSTR2 and SSTR5 have become the main biological targets of NETs with the development of somatostatin analogues (SSA). In the field of nuclear medicine, the labeling of these SSA with radionuclides allows for the synthesis of specific radiopharmaceuticals that bind to SSTRs (especially SSTR2). The use of these radiopharmaceuticals targeting SSTRs is also referred to globally as peptide receptor radionuclide therapy (PRRT).

PRRT targeting SSTRs is considered a therapeutic option in progressive SSTR-positive NETs with homogenous SSTR expression and has been rapidly incorporated on the most relevant scientific guidelines. On daily clinical practice, PRRT represents a second-line treatment option after progression to SSA on GEP-NETs with positive SSTR expression. PRRT is a molecular therapy that offers a personalized cancer treatment because radiopharmaceuticals are tailored to the unique biological characteristics of both the patient and the molecular properties of the tumor. The main goal of PRRT is to provide symptom relief, to stop or slow down tumor progression, and to improve overall survival (OS).

This review article summarizes findings published in the last years on GEP NENs, as well as unanswered clinical questions and potential future developments and directions.

## 2. Pathophysiological Basis of Radionuclide Therapy

In nuclear medicine, ionizing radiation emitted by different unstable radioisotopes to diagnose and treat diseases of different organs and systems is used.

There are three types of ionizing radiation (α, β, γ) that are characterized by their ability to penetrate matter and by the energy they transmit to it. Thus, the α and β radiations can easily be shielded by a few materials (a sheet of paper or less than an inch of material, respectively), but transmit a large amount of energy, and the γ radiations have a great penetration power and variable emission energy. In this way, radiation-emitting radionuclides α and β− will be used for treatment, while radionuclides emitting γ and β+ will be used for diagnosis [6,9] (Table 3).

Radionuclide imaging, in fact, offers the unique opportunity to detect and quantify the expression of a specific tumor biomarker through the use of a certain isotope-labeled molecules, emitting radiation suitable for imaging. Subsequently, the same radiopharmaceutical labeled with a radionuclide that emits α and β− particles to obtain a tumoricidal effect also allows for the acquisition of images that confirm the uptake of the radiopharmaceutical, the location of the lesions, and their progression over time. The whole is what is commonly known as theranostics, “we treat what we see and we see what we treat”; in a targeted and precise way, it could be considered as immunohistochemistry in vivo [10].

The radiopharmaceuticals used in this therapy have a triple structure. The principal structure is the radiometal that emits the radiation, the second is a biological vector consisting of a peptide that binds to a well-defined target (SSTR), and the third is a bifunctional chelating agent that binds the radiometal in a stable manner and allows its conjugation. A linker is usually inserted between the chelating agent and the biological vector to limit the influence of the chelating moiety. The emission of β− and α radiation by the radiometal is what allows the destruction of the NENs tumoral cells by breaking DNA strands. Analogues of the peptides with biological activity are obtained from modifications in the sequence of amino acids. For example, replacing Phe3 in the octreotide by Tyr3 (TOC) improves affinity for SSTRs (in particular SSTR2), and the introduction of a Thr (TATE) instead of Thr (ol) (TOC) further increases this affinity [9] (Figure 1).

## 3. PRRT on GEP NENs

PRRT on GEP NEN targets binds actively to the SSTR2. For more than two decades, PRRT on GEP NENs has been investigated. Compounds available include diagnostic radiotracers such as [^111^In]In-DOTA-TOC and [^90^Y]Y-DOTA-TOC (PRRT-Y), which were firstly developed, and [^177^Lu]Lu-DOTA-TATE (PRRT-Lu), which is the only PRRT approved for GEP NET patients in Europe and North America (Lutathera^®^, France). The triple structure involving this compound is the radiometal ^177^Lu, the chelating agent DOTA, and the targeting moiety octreo-TATE.

As shown in Figure 2, [^177^Lu]Lu-DOTA-TATE enters cells through its union to SSTR2. Then, it is internalized and retained in the cell lysosomes where β radiation is emitted, producing DNA chain breakdown and cell death. [^177^Lu]Lu-DOTA-TATE has a half-life of 6.7 days. As a double emitter, ^177^Lu emits both a beta and a gamma radiation. The low energy gamma radiation (0.208 MeV) enables imaging, and beta energy radiation (0.497 MeV) with a tissue penetration reaching a range of 2.2 mm (mean 0.67 mm) will enable treatment [11].

## 4. Clinical Results with PRRT-Lu on GEP NENs

The development of analogue [DOTA]0-Tyr3-octreotate or DOTATATE labeled with a dual β-γ-emitter (^177^Lu) was performed in the year 2000 [12].

### 4.1. Phase I and II Studies with PRRT-Lu

[^177^Lu]Lu-DOTA-TATE was investigated in several clinical phases I and II studies [13,14,15,16,17,18]. These studies proved an enhanced efficacy and manageability compared to previous radiopharmaceuticals, such as ^90^Y or ^111^In [1], due to a lower kidney dosimetric burden and the additional advantage of obtaining dosimetric studies and scintigraphies at the same time.

One of the most relevant of these studies was that of the Erasmus Center of Rotterdam in 2008, which evaluated toxicity, efficacy, and survival in a series of 504 patients [14]. A total of 310 patients had an advanced GEP NEN. PRRT-Lu was administered in four cycles of 7.4 GBq (GigaBecquerel) with a cumulative activity of 22.2–29.6 GBq. Median time to progression was 40 months. Median OS from the beginning of treatment was 46 months, and median OS from diagnosis was 128 months. Complete and partial remissions occurred in 2% and 28%, respectively, with minor responses in 16% and stabilization in 35%, respectively. Severe adverse events that were likely attributable to the treatment were myelodysplastic syndrome in three patients, and temporary, nonfatal liver toxicity in two patients [14].

In 2011, a phase I-II escalation dose study that aimed to define toxicity and efficacy of [^177^Lu]Lu-DOTA-TATE therapy was published. Dosage was studied dividing patients in two groups with a dosimetry-based cumulative activity up to 29 GBq. The first group received escalating activities from 3.7 to 5.18 GBq and the second from 5.18 to 7.4 GBq [15]. Progression-free survival (PFS) was 36 months, with an OS of 68% at 36 months. PRRT-Lu was well tolerated up to 29 GBq cumulative activity (up to 7.4 GBq/cycle). The maximum tolerated dose activity/cycle was not reached. No major acute or delayed renal or hematological toxicity occurred (one grade 3 leukopenia and thrombocytopenia). However, the authors recommend dividing cumulative activities into lower activity cycles.

Another phase II study was uniquely performed in patients with advanced pancreatic NETs. A total of 52 consecutive patients were treated at two different therapeutic dosages of 18.5 or 27.8 GBq in five cycles, in accordance with the patient’s kidney function and bone marrow reserve [16]. Both therapeutic dosages resulted in antitumor activity, even at a reduced total activity of 18.5 GBq. No major acute or delayed hematological toxicity occurred. Progression-free survival was significantly longer (*p* = 0.05) after a total activity of 27.8 GBq, which can thus be considered the recommended dosage in eligible patients.

On the other hand, another phase II study evaluated PRRT-Lu uniquely in patients with advanced G1 and G2 gastrointestinal NETs. A total of 43 consecutive patients with imaging progression at baseline and a positive [^111^In]In-pentetreotide scintigraphy completed treatment with [^177^Lu]Lu-DOTA-TATE. Cumulative activity was of either 18.5 or 27.8 GBq in five cycles. Both activities proved to be safe and effective in all patients. No late hematological or renal toxicity was observed in either group. PRRT-Lu was shown again to be an effective therapeutic option in advanced progressive gastrointestinal NETs with a mean PFS of 36 months [17].

A more recent study used the standard approach and described the benefit of PRRT-Lu in a standard treated group with four cycles of 7.9 GBq each one [18]. A total of 61 consecutive patients with unresectable, advanced small intestinal NET G1-2 stage IV were treated. Disease control rate was 91.8%. PFS and OS was 33 and 61 months, respectively.

The first phase II experience with PRRT-Lu in the US was reported on 2014 [19]. In this study, 37 patients with advanced grade 1 or 2 GEP NETs were treated with up to 4 cycles of [^177^Lu]Lu-DOTA-TATE to a total of 29.6 GBq. Stable disease was achieved in 41% of patients with a response rate of 28% in the entire cohort. PFS for patients who received all four cycles was 16.1 months and a total of 16.5 months for the entire cohort. No significant acute or delayed hematologic or kidney toxicity was observed.

### 4.2. Phase III Studies with PRRT-Lu: The NETTER-1 Study

NETTER-1 was the first randomized phase 3 clinical trial on PRRT [20]. A total of 231 patients with advanced, SSTR imaging-positive, midgut NETs who progressed on standard-dose octreotide LAR were randomized. Patients were either randomized to PRRT-Lu that was administered in four cycles every 8 weeks plus octreotide 30 mg every 4 weeks or to the control arm treated with high-dose octreotide LAR (60 mg every 4 weeks). Conducted with a selection of NET patients from metastatic midgut with a mean follow-up of 14 months, the study reported that treatment with [^177^Lu]Lu-DOTA-TATE determined a 79% reduction in risk progression or death compared to high doses of octreotide (*p* > 0.001, HR 0.21). The response rate in the group of PRRT-Lu was 18% versus 3% in the control group. Estimated PFS with PRRT-Lu was of 40 months compared to 8.4 months with high-dose SSA. These markedly higher PFS results on PRRT-Lu arm is what led to the Food and Drug Administration (FDA) and European Medicines Agency (EMA) approval. Consecutively, the most relevant clinical guidelines incorporated this treatment into their respective algorithms [21,22,23,24,25,26].

Moreover, the NETTER-1 study [27] demonstrated the excellent quality life of patients treated with [^177^Lu]Lu-DOTA-TATE in different aspects such as general health, body image, functionality (general and occupational), diarrhea, pain, fatigue, and worry about the disease. An analysis of time up to quality-of-life deterioration in both arms of the study was shown to be much higher in patients with PRRT-Lu compared to the control group (28.8 months for PRRT-Lu versus 6.1 with SSA) with a RR of 0.40.

After the 5 year follow-up, recent data on OS have been published [28]. Of the totality of randomized patients, 86.3% in the PRRT-Lu arm and 86.8% in the control arm entered long-term follow-up (*n* = 200). Median OS in the control arm was 36.3 months, and 40 months in the PRRT-Lu arm (HR 0.84 (95% CI: 0.60, 1.17) with *p* = 0.30). During follow-up, there was a 36% of patient cross-over to the PRRT arm, and this was the case in the majority of the patients within 24 months. Although the difference on OS was of 11.7 months, this was not statistically significant, which was attributed by the investigators to the impact of a high-rate cross-over of patients in the control arm to PRRT after progression.

Concerning adverse events, 1.8% PRRT-Lu patients in the study developed myelodysplastic syndrome, but no new cases of myelodysplastic syndrome or acute leukemia were reported in the long-term follow-up.

Therefore, the NETTER-1 study demonstrated that PRRT-Lu yielded a statistically and clinically significant improvement in PFS as a primary endpoint (HR: 0.18, *p* < 0.0001), although improvement in OS was not demonstrated, probably due to a high cross-over in the control arm.

### 4.3. Meta-Analysis

A meta-analysis undertaken in 2015 included six studies with a total of 473 GEP NENs, and, although the treatment protocols were not standardized and the treatment effects should be further verified through prospective randomized controlled trials, the authors concluded that PRRT-Lu is an effective option for patients with inoperable or metastatic NENs achieving a disease control rate of approximately 80% [29].

### 4.4. Cohort Studies with PRRT-Lu

Although hierarchy of evidence to guide clinical interventions is less with case–control and cohort studies, several large well-designed cohort studies of PRRT-Lu have been reported that contribute to adding more information of this treatment and its safety in patients with GEP NET.

One of these cohort studies was the one performed in a Dutch population—610 patients were included with GEP and bronchial NETs [30]. A total of 443 of these patients were treated with a cumulative activity of at least 22.2 GBq before 2013. PFS and OS for all patients were 29 months and 63 months, respectively. Long-term toxicity included myelodysplastic syndrome in 1.5% and acute leukemia in four patients 0.7%. No therapy-related long-term hepatic or kidney failure were reported.

A cohort analysis of 1048 patients with NETs who underwent either PRRT-Lu, PRRT-Y, or both therapies in an alternating fashion has been reported [31]. In total, 74% of these patients had GEP NETs. The best OS was achieved by a combination of PRRT-Y and PRRT-Lu. The shortest survival was observed in patients treated exclusively with PRRT-Y, while PRRT-Lu alone resulted in an intermediate survival. Median OS in the PRRT-Lu patients was 44 months, while median PFS was 17 months. Patients treated with both therapies had an OS of 64 months with a median PFS of 24 months.

A very relevant study that opens a window to the use of [^177^Lu]Lu-DOTA-TATE in patients with G3 NETS is a smaller cohort study that reported the experience of 69 G3 NET patients. Patients either received PRRT-Lu or PRRT-Y [32]. The clinical outcome was promising, especially in patients with a Ki-67 index of less than or equal to 55% with a median PFS of 11 months and OS of 22 months. This outcome was observed even in patients for whom chemotherapy had failed.

## 5. Position Statement of PRRT in Treatment of GEP NENs

As previously commented, the phase I and II studies, cohort studies, and especially the NETTER 1 study yielded very relevant results that made scientific societies incorporate PRRT-Lu on their clinical guidelines [21,22,23,24,25,26] (Table 4). It was approved in September 2017 in Europe by the EMA and in January 2018 by the FDA as the first radiopharmaceutical for PRRT in progressive gastroenteropancreatic NET.

[^177^Lu]Lu-DOTA-TATE is considered a therapeutic option in progressive SSTR-positive NETs with homogenous SSTR expression [33]. Eligibility and clinical decision making should be based on multidisciplinary discussion [34]. European Neuroendocrine Tumor Society (ENETS) consensus guidelines consider the most important inclusion criteria: inoperable/metastatic well-differentiated (G1/G2) NET, sufficient tumor uptake on the diagnostic SSTR functional images, sufficient bone marrow reserves, creatinine clearance > 50 mL/min, expected survival >3 months, and Karnofsky performance status >50. Well-differentiated G3 NET may as well be considered, although further data are required on response rates and survival [34]. Until the moment there are no randomized clinical trials available that compare optimal administered activity per treatment cycle, timing between cycles, or adequate cumulative administered activity for PRRT-Lu, the recommended posology is of up to four cycles of a fixed activity of 7.4 GBq per cycle, according to those used on NETTER-1 clinical trial that, in turn, is based on the protocol developed at the Erasmus Medical Center [20,35].

Where and how to place PRRT on the sequencing of treatment on NETs after progression to SSA still raises many doubts. ENETS consensus guidelines published in 2016 place PRRT-Lu as a possible treatment in second line after SSA along with everolimus, locoregional therapies, and IFN alpha 2b in advanced midgut NETs. The sequencing of PRRT-Lu as second- or third-line therapy for advanced intestinal NET also depends on other issues, including accessibility to PRRT. A strong SSTR expression on imaging is necessary to achieve better results with PRRT, while extensive bone and/or liver disease as well as a deteriorated kidney function may limit its use. PRRT-Lu may be therefore recommended in midgut NET as a second-line therapy after failure of SSA if the general requirements for applying PRRT are fulfilled or as a third-line therapy after failure of everolimus. In advanced pancreatic NETs, these guidelines position PRRT-Lu in G1/G2 NET after failure of medical therapy including SSA, chemotherapy, or novel targeted drugs (sunitinib or everolimus). This positioning in pancreatic NETs is due to the lack of a prospective trial with PRRT-Lu in pancreatic NET; however, the authors consider that a potential increasing toxicity used after prior chemotherapy or targeted therapy might justify an earlier use of PRRT-Lu in selected patients [36].

European Society of Medical Oncology (ESMO) guidelines [37], recently published in 2020, agree with ENETS guidelines that PRRT-Lu can be recommended in patients with midgut NETs and disease progression to SSAs. As well as in pancreatic NETs in which these guidelines coincide that since randomized trials are lacking in this subgroup of patients, molecular target therapies should be a prior treatment choice before PRRT-Lu. The main subtle difference in these guidelines compared to the previous ones regarding PRRT is that levels of evidence are incorporated.

North American Neuroendocrine Tumor Society (NANETS) guidelines [38] also published in 2020 coincide mainly with the previous ones with two subtle differences. The first difference is on midgut NETs, in which a differentiation between functioning and non-functioning is made. While on functioning midgut, PRRT is recommended as a second line, on non-functioning midgut, PRRT could be recommended as second- or third-line therapy after everolimus. The other peculiarity is that it includes the option of using PRRT as a first-line therapy option in patients with very high tumor burden where any further growth would entail significant risk.

## 6. Unanswered Questions on PRRT in Patients with GEP NENs

Although treatment PRRT yields promising results in patients with GEP NENs regarding clinical response and low toxicity, many questions remain unanswered, such as the following: Which is the correct dose and individual dosimetry? Which is the place for salvage PRRT-Lu? Which is the role of PRRT-Lu in the pediatric population? Which is the optimal sequencing of [^177^Lu]Lu-DOTA-TATE in advanced GEP NETs? Which is the place of [^177^Lu]Lu-DOTA-TATE in G3 NENs?

### 6.1. Which Is the Correct Dose and Individual Dosimetry?

As mentioned previously, up until now, there are no randomized clinical trials that compare optimal activity per cycle, timing between cycles, or cumulative administered activity for [^177^Lu]Lu-DOTA-TATE. PRRT-Lu is therefore administered in four cycles of a fixed activity of 7.4 GBq per cycle. This fixed activity results in variable absorbed doses both in healthy organs and in the tumor. There is a large patient variability in the biodistribution of radiopharmaceuticals [39]. A special emphasis should be made on the absorbed doses for bone marrow and kidney, since they are considered as the dose-limiting organs in [^177^Lu]Lu-DOTA-TATE [40,41]. Personalized medicine achieving an individual dosimetry-based activity administration could help optimizing treatment and reducing secondary effects. Methods for accurate tumor and normal tissue dosimetry and determination of predictive factors that are associated with higher uptake of the radiopharmaceutical in NETs are needed [42].

Dosimetry can be based on pre-PRRT with [^68^Ga]Ga-DOTA-SSA PET/CT or post-PRRT single-photon emission computed tomography (SPECT) scanning.

The group from Uppsala evaluated the dose–response relationship for pancreatic NETs treated with PRRT-Lu. The absorbed dose calculations relied on sequential post-treatment distribution SPECT/CT imaging at 24, 96, and 168 h after infusion of [^177^Lu]Lu-DOTA-TATE [43]. Tumor-absorbed doses necessary to reach best response ranged from 10 to 340 Gy. The results implied a significant correlation between absorbed dose and tumor reduction; however, large variations in response for similar absorbed doses were observed. Previously, this same group described that in 50% of patients, more than four cycles of 7.4 GBq could be administered, and patients were treated with up to 10 cycles [44].

Another group performed dosimetry on the basis of the kidney-absorbed dose. The aim of this study was to assess the accuracy and inter-observer reproducibility of simplified dosimetry protocols on the basis of quantitative two-time-point SPECT (QSPECT), which provided reproducible and accurate dose estimates. The authors affirm that the kidney-absorbed dose over the four-cycle induction PRRT course can be standardized by personalizing injected activity on the basis of the product of glomerular filtration rate with lean body weight or body surface area for the first cycle and on prior renal dosimetry for the subsequent cycles. The results were encouraging without adding an extra toxicity [45].

Monte Carlo simulation is a non-deterministic or numerical statistical method, used to approximate complex mathematical expressions that are costly to evaluate accurately. By introducing the map of voxel-wise, time-integrated activity or mass density distribution into the Monte Carlo simulation, one can estimate the related absorbed energy dose distribution in an interesting target region [46]. Recently, a novel Monte Carlo-based voxel-wise dosimetry approach to determine organ- and tumor-specific total tumor doses has been published. After the first cycle of therapy, regions of interest were defined manually on the SPECT/CT images for the spleen, the kidneys, and all tracer-positive tumor lesions. Four SPECT images, taken at 4 h, 24 h, 48 h, and 72 h after injection of [^177^Lu]Lu-DOTA-TOC, were used to determine their effective half-lives in the structures of interest. The absorbed doses were calculated by a three-dimensional dosimetry method based on Monte Carlo simulations. Total tumor doses were calculated as the sum of all products of single tumor doses with single tumor volumes divided by the sum of all tumor volumes. These results were promising as a new tool to predict the total tumor dose received [42].

Ongoing trials aim to prove the feasibility and efficacy of the individual dosimetry-based strategy for PRRT improvement. One of these trials is ClinicalTrials.gov Identifier: NCT04467567, in which the use of a new camera, VERITON-CT™ CZT (Spectrum Dynamics Medical, Morges, Switzerland), is being explored. The primary outcome is the comparison of dosimetric results obtained from a conventional gamma camera and the new VERITON-CT™ CZT camera. Secondary outcomes include both measurement of average absorbed doses in tumor lesions and in limiting organs. Correlation between the dosimetry result and average absorbed doses to the kidneys and to the bone marrow versus renal function are studied [47].

Another trial is aiming to evaluate safety and dosimetry of PRRT-Lu in adolescent patients with GEP-NETs and paragangliomas (ClinicalTrials.gov Identifier: NCT04711135). The dosimetric study will be performed during the first week after the first treatment dose, which will allow an estimation of the cumulative absorbed radiation dose and for decision making on the following dose levels.

### 6.2. Which Is the Place for Salvage PRRT-Lu?

Eventually, most patients with advanced GEP NETs will present progressive disease after PRRT. The efficacy and safety of PRRT salvage treatment (R-PRRT) have been analyzed in studies with small patient cohort with NETs at different locations and stages and with limited follow-up. In some studies, different radiopharmaceuticals or their combinations have been used in initial PRRT and salvage treatment. In these studies, there was a great variability in the number of cycles as well as in the cumulative activities administered [48,49,50,51,52,53,54].

The largest cohort of patients from a single institution who underwent initial PRRT-Lu and R-PRRT-Lu, with the longest follow-up, is that of the Erasmus Center [49]. This study included R-PRRT-Lu in 168 GEP and bronchial NET patients. Patients received a median cumulative activity dose of 14.9 GBq (3.7–16.2 GBq), resulting in total median cumulative administered activities doses of 44.7 GBq (26.3–46.4 GBq) after retreatment and 59.7 GBq (55.2–60.5 GBq) after re-retreatment. Patients with bronchial NET or GEP-NET with R-PRRT-Lu had a significantly longer OS than patients in the control group. Patients with midgut NET had an OS was 77.3 months compared to 51 months in the control group. Patients with pancreatic NET had an improved OS following R-PRRT: OS was 93.9 months in the retreatment group and 61.5 months in the control group.

Recently, two meta-analyses and systematic reviews have been published [55,56]. Both studies agree that R-PRRT may be a therapeutic choice in patients with progressive GEP NETs. The first of these studies [55] evaluated a total of 414 patients—median PFS was 12.52 months (95% CI: 9.82–15.22) and a median OS was 26.78 months (95% CI: 18.73–34.83). The second [56] evaluated a total of 426 patients. R-PRRT raised an estimated PFS of 14.1 months (95% CI: 12.2–15.9) and OS of 26.8 months (95% CI: 18.8–34.9). Both agree that R-PRRT-Lu provided encouraging PFS in patients and a safety profile similar to initial PRRT.

However, concerns about which is the appropriate time for R-PRRT after progression (12–18 months), long-term toxicities in retreated patients, and its role in G2-G3 patients exist. These unresolved issues may be due to the lack of well-designed studies that would definitively clarify these clinical and safety questions encountered with R-PRRT.

### 6.3. Which Is the Role of PRRT-Lu in Pediatric Population?

Although overall incidence and prevalence of GEP-NETs has been gradually increasing over the last few decades, data on the pediatric population specifically are limited due to its rare occurrence in this population [57]. NETs in children show a high SSTR expression and could be potentially treated with PRRT-Lu; however, children under the age of 18 have been excluded from participation on PRRT trials.

Data regarding its use in other pediatric tumors such as neuroblastoma and medulloblastoma, which also express SSTR 2, have been published with promising results. Recently preliminary studies showed that PRRT-Lu is tolerable and feasible in pediatric patients with SSTR2-positive tumors. However, long-term efficacy and adverse effects in children are unknown [58].

In children, there is no widespread experience in the treatment with PRRT-Lu, and the injected activities should be adapted per square meter [59,60]. In general, the same activity is used as in adults adapted to the weight with a maximum activity of 7.4 Gbq per infusion, with a total of four cycles every 8 weeks [58].

This unresolved question opens the need for further clinical trials PRRT-Lu in the pediatric population with GEP-NETs in which dosimetry data should be collected as well. The recent, abovementioned clinical trial NCT04711135 will explore some of these issues.

### 6.4. Which Is the Optimal Sequencing of PRRT-Lu in Advanced GEP NETs?

As above mentioned, treatment with [^177^Lu]Lu-DOTA-TATE is placed as a second-line therapy along with everolimus in midgut NET tumors and as third or fourth line in pancreatic tumors. Despite the lack of data from specific phase III clinical trials, there is positive evidence of efficacy and safety in advanced pancreatic NET patients in which PRRT could surpass on treatment sequencing other targeted therapies [61,62]. Moreover, a potential increasing toxicity used after prior chemotherapy or targeted therapy has been described, which might justify an earlier use of PRRT in selected patients [36].

Ongoing, well-designed randomized controlled trials (RCT) could enhance the most appropriate sequencing in patients with advanced metastatic NETs after treatment with SSA.

One of these RCTs is the ClinicalTrials.gov Identifier: NCT03049189 [63]: Efficacy and Safety of PRRT-Lu in GEP-NET Patients (COMPETE). The COMPETE trial is a prospective, randomized, controlled, open-label, multicenter phase III study to evaluate efficacy and safety of PRRT-Lu compared to targeted molecular therapy with everolimus in patients with inoperable, progressive, SSTR-positive (SSTR+) GEP NET.

Another of these RCTs is the ClinicalTrials.gov Identifier: NCT02230176 [64]: antitumor efficacy of PRRT-Lu randomized vs. sunitinib in unresectable progressive well-differentiated pancreatic NET: first randomized phase II (OCCLURANDOM). The OCCLURANDOM trial is the first randomized, open-label, national, multicenter, phase II study assessing the efficacy and safety of OCLU in subjects with pretreated progressive pancreatic, inoperable, SSTR+, well-differentiated pancreatic NET. Subjects must have experienced documented progression of disease within 1 year prior to the start of the study. The control group of patients receiving sunitinib will be used as internal control to assess the hypothesis of 12 months PFS equal to 35% in patients receiving sunitinib.

### 6.5. Which Is the Place of PRRT-Lu in G3 NENs?

SSTR density expression on GEP NETS according to WHO classification shows that G2/G3 NETs have a considerable expression (Table 1), and therefore PRRT has been considered as a therapeutic option in these group of patients.

Published data in these subgroups of patients are scarce, and studies are retrospective [32,65,66,67]. The published data are of 280 NEN G3 patients, and the overall results show PFS between 9 and 23 months and an OS between 19 and 53 months. Disease control rates range between 30 and 80%. Patients with lower Ki-67 proliferation rate (<55%) show better response compared to those with higher values.

These preliminary results have made recent ESMO guidelines to include PRRT-Lu as a therapeutic option in NET G3 patients with a level of evidence of IV and grade C of recommendation [37]. New RCTs have been recently initiated to address this issue.

The NETTER-2 (ClinicalTrials.gov Identifier: NCT03972488) trial [68] is a phase III multi-center, randomized, open-label study to evaluate the efficacy and safety of [^177^Lu]Lu-DOTA-TATE in patients with grade 2 and grade 3 advanced GEP-NETS. The aim of NETTER-2 is to determine if PRRT-Lu in combination with long-acting octreotide prolongs PFS in GEP-NET patients with high proliferation rate tumors (G2 and G3) when given as a first-line treatment compared to treatment with high-dose (60 mg), long-acting octreotide.

The COMPOSE (ClinicalTrials.gov Identifier: NCT04919226) trial, which has not yet started recruitment [69], is a prospective, randomized, controlled, open-label, multicenter study to evaluate efficacy, safety, and patient-reported outcomes of PRRT-Lu compared to best standard of care in patients with well-differentiated aggressive grade 2 and grade 3, SSTR+, GEP NETs. PFS and OS are the primary and secondary outcomes, respectively, and PRRT will be compared with the best standard of care in these patients, which includes treatment with everolimus, FOLFOX, and CAPTEM.

## 7. Future Investigations on PRRT in Patients with GEP NENs

Different methods to improve the objective response rate and survival are under investigation. This includes changes of the radionuclide or radiopharmaceutical, combination therapy with different agents such as radiosensitizers, or other abovementioned issues such as dosimetry (Figure 3).

### 7.1. New Radionuclides

The use of alpha-emitting radionuclides in PRRT could potentially increase the level of tumoral cell death by increasing the number of DNA double-strand breaks [70]. These radionuclides include ^213^Bismuth (^213^Bi), ^225^Actinium (^225^Ac), and ^212^Lead (^212^Pb). The literature with these radionuclides is scarce, and there is a need for good clinical trials regarding the dose-limited toxicity and efficacy.

Promising results with ^213^Bi [71] have been published. Enduring responses were observed in all treated patients. Chronic kidney toxicity was moderate. Acute hematotoxicity was even less pronounced than with the preceding beta therapies.

Recently, in July 2021, a phase 1 study was completed (ClinicalTrials.gov Identifier: NCT03466216) in which [^212^Pb]Pb-DOTAM-TATE was tested [72]. This drug substitutes beta emitters currently being used (i.e., ^177^Lu or ^90^Y) with an alpha emitter (^212^Pb). The reason for the change to this alpha emitter is that it will provide significantly higher linear energy transfer, causing more tumor death and a short penetration depth, which would result in less collateral damage in healthy tissue. The primary objective is to assess the safety and dose-limiting toxicity using ascending doses of this alpha emitter. The secondary objectives are to determine the pharmacokinetic properties and preliminary effectiveness (of AlphaMedix™, Petach Tikva, Israel). Results from this study are not yet published.

### 7.2. Somatostatin Antagonists

Up until now, SSA have been used, as it was believed that receptor internalization was essential for a higher tumor uptake. However, toxicity remains the main concern with these agents since an increase uptake in bone marrow or kidney has been described [70].

However, preclinical studies in mice have found a significantly higher tumor uptake with somatostatin antagonist as well as a longer survival and better tumor control [73,74]. Compared to the agonist [^177^Lu] Lu-DOTA-TATE, the antagonist [^177^Lu] Lu-DOTA-JR11 showed 1.7–10.6 times higher tumor dose uptake in humans [75]. These promising results have been explored on an interventional study in which experimental imaging with [^68^Ga]Ga-DOTA-JR11 and experimental treatment with [^177^Lu]Lu-DOTA-JR11 are to be explored. The primary outcome is overall response rate according to RECIST 1.1 and median PFS and OS [76].

### 7.3. Upregulation of SSTR2 Expression

SSTR2 is the highest affinity target used for imaging and treatment in NETs. Strategies to improve binding of somatostatin agonists to this receptor are being studied by introducing epigenetic modulation of the SSTR2 gene. This epigenetic modulation has been induced by epidrug treatment such as valproic acid [77,78], and more recently by epidrugs 5-aza-2′-deoxycytidine and valproic acid [79]. This last group showed an increased uptake of radiolabeled octreotide, as well as increased sensitivity to the SSA octreotide in functional cAMP inhibition. At epigenetic level, they observed a low methylation level of the SSTR2 gene promoter region, irrespective of expression. Activating histone mark H3K9Ac could be regulated with epidrug treatment. The conclusion of this study is that epidrug treatment might hold promise for improving and adding to current somatostatin agonist treatment strategies of patients with pancreatic NETs.

### 7.4. Combined Therapies to Improve PRRT Efficacy

In order for PRRT outcomes to be further improved, the use of combination therapies is being attempted at different cellular levels [80]: increasing cellular DNA damage with traditional chemotherapies, radio sensitization inhibiting DNA repair, radio sensitization inhibiting phosphoinositide-3-kinase/protein kinase b/mammalian target of rapamycin (mTOR) signaling, radio sensitization inhibiting hedgehog signaling, radio sensitization inhibiting p53–murine double minute 2 (MDM2) interactions, radio sensitization disrupting cell cycle, radio sensitization disrupting nicotinamide adenine dinucleotide (NAD1) metabolism, and radio sensitization blocking immune checkpoints (Table 5) [80,81,82,83,84,85,86,87,88,89,90,91,92,93,94,95,96,97,98,99,100,101,102,103,104,105,106,107,108,109].

## 8. Conclusions

PRRT-Lu has undoubtedly come to stay as a cornerstone treatment in metastatic GEP-NET. PRRT-Lu has been demonstrated to be a treatment option that is highly effective in controlling advanced, metastatic, or inoperable progressive neuroendocrine tumor disease. PRRT is rarely curative but has been shown to help relieve symptoms, improve a patient’s quality of life, shrink tumor lesions, and slow the progression of the disease. PRRT-Lu is considered as a second line therapy in midgut NET and on further line therapies on pancreatic NET by the most relevant scientific societies (ENETS, ESMO, and NANETS). However, many questions still remain unresolved, and future directions are moving towards optimizing dosimetry; the use of somatostatin antagonists; combined therapies that increase tumoral DNA damage such as capecitabine or temozolomide; or inhibiting tumoral DNA repair such as PARP, HSP90, or topoisomerase inhibitors, as well as other cellular signaling inhibitors, at the same time as repositioning of treatment on algorithms or its use on GEP-NET G3, which will surely be answered in the upcoming clinical trials.

## Figures and Tables

**Figure 1 cancers-14-00584-f001:**
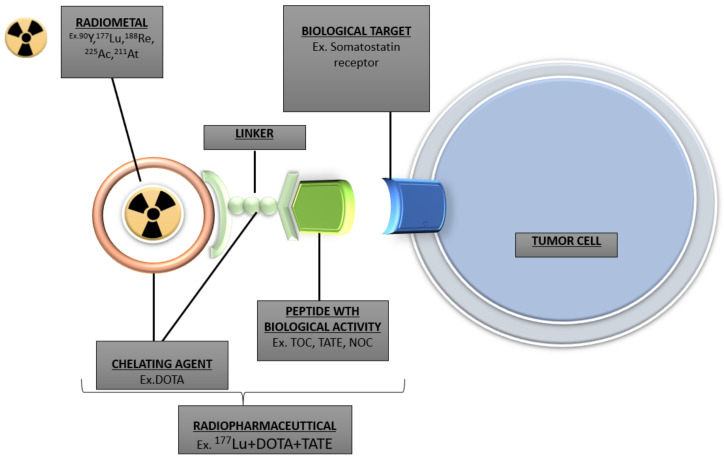
Schematic design of radiopharmaceutical complex.

**Figure 2 cancers-14-00584-f002:**
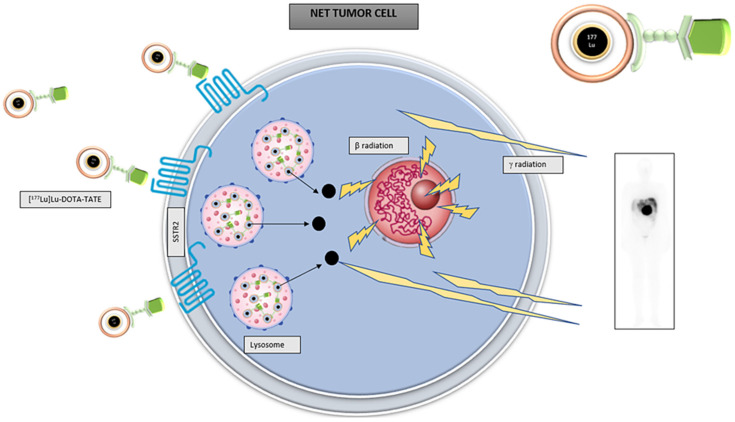
Schematic design of [^177^Lu]Lu-DOTA-TATE action over NET cells.

**Figure 3 cancers-14-00584-f003:**
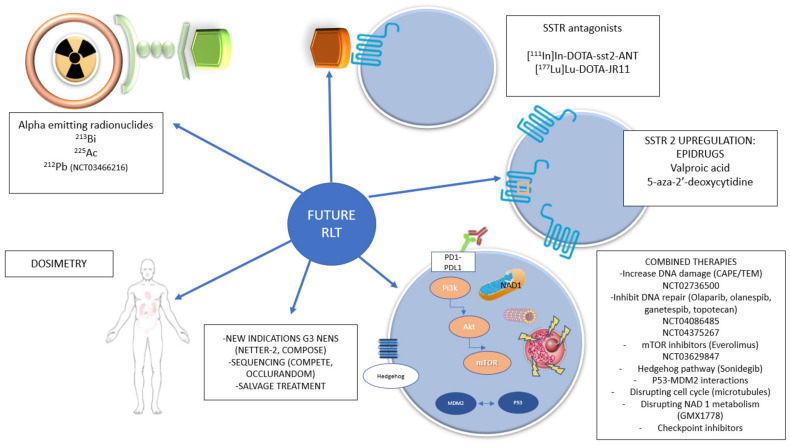
Schematic design of future directions on RLT.

**Table 1 cancers-14-00584-t001:** Somatostatin density receptor expression on GEP NETS according to WHO classification.

WHO Classification	SSTR1 (%)	SSTR2 (%)	SSTR3 (%)	SSTR4 (%)	SSTR5 (%)
G1	80–100%	80–100%	30–80%	<15%	80–100%
G2	80–100%	80–100%	30–80%	<15%	30–80%
G3	80–100%	80–100%	30–80%	<15%	<15%

**Table 2 cancers-14-00584-t002:** Somatostatin density receptor expression according to primary site location and/or functional GEP NETS.

Primary Site Location	SSTR1	SSTR2	SSTR3	SSTR4	SSTR5
Pancreas	<15%	80–100%	<15%	<15%	30–80%
Gastric	<15%	80–100%	<15%	<15%	30–80%
Intestinal	<15%	30–80%	30–80%	<15%	<15%
Insulinoma	15–30%	30–80%	30–80%	<15%	30–80%
Gastrinoma	15–30%	80–100%	30–80%	<15%	30–80%

**Table 3 cancers-14-00584-t003:** Main radionuclides used for imaging and therapy according to their type of radiation.

Application		Radionuclide	Type of Emission	Source
Imaging	SPECT	^99m^Tc	γ	Generator
		^111^In	γ	Cyclotron
	PET	^68^Ga	β+	Generator/Cyclotron
		^18^F	β+	Cyclotron
		^64^Cu	β+/β−/γ	Cyclotron
Therapy		^90^Y	β−	Generator
		^177^Lu	β−/γ	Cyclotron
		^188^Re	β−/γ	Generator
		^225^Ac	α	Generator
		^211^At	α	Cyclotron

Single photon emission computed tomography (SPECT); positron emission tomography (PET).

**Table 4 cancers-14-00584-t004:** Position statement on PRRT of the different scientific societies.

Scientific Guideline	Year of Publication	Inclusion Criteria for Treatment	Sequencing on Pancreatic NET	Sequencing Midgut NET	Other Tumors or Circumstances That May Be Considerated for Treatment
ENETS	2016	− Inoperable/metastatic well-differentiated (G1/G2) NET − Tumor uptake SSTR functional images − Homogenous SSTR expression (all NET lesions are positive) − Bone marrow reserves Creatinine clearance >50 mL/min − Expected survival > 3 months − Karnofsky performance status >50	− Third or subsequent lines. PRRT after failure to other therapies: SSA, sunitinib, everolimus, or chemotherapy	− Second line option after SSA Or − Third line after everolimus	− Well-differentiated G3 NET
ESMO	2020	− ENETS criteria	− Third or subsequent lines. PRRT after failure to other therapies: SSA, sunitinib, everolimus, chemotherapy. − Level of evidence III A	− Second line option after SSA Or − Further lines − Level of evidence I A	− NET G3 − Level of evidence IV C
NANETS	2020	− SSTR-positive tumors	− Third or subsequent lines. PRRT after failure to other therapies: SSA, everolimus, sunitinib, and chemotherapy	− Functional midgut: Second line after SSA − Nonfunctional midgut: could be second line or third line after everolimus	− Could be considered a first-line option in patients with very high tumor burden where any further growth would entail significant risk

**Table 5 cancers-14-00584-t005:** Main clinical and preclinical studies performed or currently underway of combined therapies to improve PRRT efficacy on NETs.

Cellular Level	Agents	Clinical Studies	Preclinical Studies
Increasing cellular DNA damage	[^177^Lu]Lu-DOTA-TATE + capecitabine	[81,82,83,84] NCT02736500 [8]	[85]
	[^177^Lu]Lu-DOTA-TATE + temozolomide	[86]	
	[^177^Lu]Lu-DOTA-TATE + capecitabine + temozolomide	[86,87,88,89,90]	
Inhibiting DNA repair	PARP inhibitors: olaparib	NCT04086485 [94] NCT04375267 [95]	[91,92,93]
	HSP90 inhibitors: onalespib and ganetespib	-	[96,97]
	Topoisomerase inhibitors: topotecan	-	[98]
Inhibiting phosphoinositide-3-kinase/protein kinase b/mammalian target of rapamycin (mtor) signaling	Everolimus	[104] NCT03629847 [105]	[99,100,101,102,103]
Inhibiting hedgehog signaling	Sonidegib	-	[106]
Inhibiting p53–murine double minute 2 (MDM2) interactions	-	-	[107]
Disrupting cell cycle/microtubules	Taxanes	In prostate cancer	-
Disrupting nicotinamide adenine dinucleotide (NAD1) metabolism	Inhibitor GMX1778	-	[108]
Blocking immune checkpoints	Nivolumab	[109] (Lung NET)	-

Poly-ADP ribose polymerase inhibitors: PARP inhibitors.

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
