# Peer review of "Peptide Receptor Radionuclide Therapy with [177Lu]Lu-DOTA-TATE in Patients with Advanced GEP NENS: Present and Future Directions"

_cancers, 2022, doi:10.3390/cancers14030584_

Round 1

Reviewer 1 Report

Section 2:

Table 3.  Re188 (188W/188Re generator) production is referred to cyclotron

94-98 rows: in the radiopharmaceutical description: the presence and the role of a spacer linker as well as the modification of the Tyr3-octreotate are omitted, with particular reference to the affinity changes for receptor subtypes.

Section 5.

I suggest to include a table which summarize the position of scientific associations about the role of PRRT

Reviewer 2 Report

The manuscript of del Olmo-Garcia describes the use of peptide-receptor radionuclide therapy for patients bearing GEP-NETS tumors. While the content is interesting the manuscript is poorly written, especially in the introduction/conclusion. The manuscript also lacks good reviewing from its authors as seen by the huge number of typos (lot more of what is expected of a manuscript submitted in a journal such as Cancers). Few examples below (not limited to):

  • the writing of radiopharmaceuticals is not homogenized in the entire manuscript: the isotope should be in exponent and radiopharmaceuticals should be written always in the same manner, i.e. using same international nomenclature as for [177Lu]Lu-DOTA-TATE without space between [177Lu]Lu. Please homogenize in text and figures.
  • FDA/EMA sometimes written entirely, sometimes abbreviated, please homogenize.
  • Typos in figures, examples: “Emmitting” has been written instead of “emitting”.
  • L351-352: repetition of the same sentence.
  • Several abbreviations are not described neither in the text nor in tables/figures and can thus induce misunderstanding for readers.

L366 “Recently, a novel Monte-Carlo-based voxel wise dosimetry” It could be more relevant to introduce first what is a dosimetry -based on Monte-Carlo and Factor S calculations before explaining that a “new” method has been used.

Paragraph L430-433 should be rewritten. Ci and Bq are NOT doses, these are only injected activities.  

MBq/GBq is the international unit for injected activities, there is no need to indicate the corresponding amount in mCi in parenthesis as those should not been used anymore since several years now L431-342 and in few other sentences over the manuscript.

L92: the common English term for a radionuclide which is both used for imaging and therapy is “Theranostic”, not “teragnosis”.

L366 “Recently, a novel Monte-Carlo-based voxel wise dosimetry […]” It could be relevant to introduce first what is a dosimetry -based on Monte-Carlo and Factor S calculations before explaining that a “new” method has been used.

L61-62: “[…] SSTR has become the target for the development of specific radionuclides”. Here it concerns the development of radiopharmaceuticals, not radionuclides. L62 “Such radionuclides…” same here. Paragraph L59-63 is unclear and should be rewrite.

Reviewer 3 Report

Thank you for a very well-written and very interesting manuscript. 

Only minor comments

  1. The special issue is targeted NET specialists globally. Therefore, I recommend to remove the section about the Spanish guidelines (SEOM).
  2. The p-value with respect to OS in the NETTER-1 paper from 2021 was 0.3. The difference is not even close to statistically significant, therefore I would recommend to rephrase the term "clinical trend". 
  3. At page 8, line 333 please rephrase "until the moment". 

Round 2

Reviewer 2 Report

The quality of the article has been greatly improved and should now meet the requirements for publication.